# Diagnostic Accuracy of a Portable ECG Device in Rowing Athletes

**DOI:** 10.3390/diagnostics12102271

**Published:** 2022-09-20

**Authors:** Fiona Wilson, Cliodhna McHugh, Caroline MacManus, Aaron Baggish, Christopher Tanayan, Satyajit Reddy, Meagan M. Wasfy, Richard B. Reilly

**Affiliations:** 1Discipline of Physiotherapy, School of Medicine, Trinity College Dublin, D08 W9RT Dublin, Ireland; 2Discipline of Physiology, School of Medicine, Trinity College Dublin, D02 R590 Dublin, Ireland; 3High Performance Sport New Zealand, Auckland 0632, New Zealand; 4Cardiovascular Performance Programme, Massachusetts General Hospital, Boston, MA 02114, USA; 5Centre for Bioengineering, School of Medicine, Trinity College Dublin, D02 R590 Dublin, Ireland

**Keywords:** electrocardiogram, portable device, athletes, arrythmia, sports cardiology

## Abstract

Background: Athletes can experience exercise-induced transient arrythmias during high-intensity exercise or competition, which are difficult to capture on traditional Holter monitors or replicate in clinical exercise testing. The aim of this study was to investigate the reliability of a portable single channel ECG sensor and data recorder (PluxECG) and to evaluate the confidence and reliability in interpretation of ECGs recorded using the PluxECG during remote rowing. Methods: This was a two-phase study on rowing athletes. Phase I assessed the accuracy and precision of heart rate (HR) using the PluxECG system compared to a reference 12-lead ECG system. Phase II evaluated the confidence and reliability in interpretation of ECGs during ergometer (ERG) and on-water (OW) rowing at moderate and high intensities. ECGs were reviewed by two expert readers for HR, rhythm, artifact and confidence in interpretation. Results: Findings from Phase I found that 91.9% of samples were within the 95% confidence interval for the instantaneous value of the changing exercising HR. The mean correlation coefficient across participants and tests was 0.9886 (σ = 0.0002, SD = 0.017) and between the two systems at elevated HR was 0.9676 (σ = 0.002, SD = 0.05). Findings from Phase II found significant differences for the presence of artifacts and confidence in interpretation in ECGs between readers’ for both intensities and testing conditions. Interpretation of ECGs for OW rowing had a lower level of reader agreement than ERG rowing for HR, rhythm, and artifact. Using consensus data between readers’ significant differences were apparent between OW and ERG rowing at high-intensity rowing for HR (*p* = 0.05) and artifact (*p* = 0.01). ECGs were deemed of moderate-low quality based on confidence in interpretation and the presence of artifacts. Conclusions: The PluxECG device records accurate and reliable HR but not ECG data during exercise in rowers. The quality of ECG tracing derived from the PluxECG device is moderate-low, therefore the confidence in ECG interpretation using the PluxECG device when recorded on open water is inadequate at this time.

## 1. Introduction

Monitoring human physiologic function and performance during real-time activities is an important part of sport and exercise medicine/science [1]. Electrophysiological data is now widely collected with sophisticated wearable devices that are constantly improving in technology. These include heart rate (HR) sensors, accelerometers, plethysmographs and global positioning systems [2]. These devices collect electrophysiological data from athletes to optimise performance, reduce injury and increase safety [3,4]. Specific to cardiac function, the ambulatory electrocardiogram (ECG) is commonly employed to detect cardiac arrhythmias outside a clinic setting.

Competitive rowing, an endurance-based sport, involves abrupt transitions between rest and high-intensity exercise [5]. There are distinct differences in aerobic and cardio-respiratory demands between ergometer (ERG) rowing and on-water (OW) rowing [5,6]. There is an increased risk for middle-aged male endurance athletes to develop atrial fibrillation (AF) [7,8]; thus, monitoring of rowing athletes is appropriate, particularly with the rapid expansion of participation in masters rowing training and competing. Current devices available for monitoring athletes’ ECGs include short-term rhythm monitors, such as two-week patches and “ad hoc” monitors such as Kardia and Apple watches. However, these devices are limited due to a lack of reusability (two-week patches) and the inability to measure ECG tracing while athletes are engaged in exercise as athletes are required to stop the exercise to activate the device. Specific to rowing athletes, research assessing ECG characteristics during OW rowing is lacking. It is unclear if ECG characteristics produced during ERG rowing directly reflect those of OW rowing. Therefore, an ability to obtain accurate ECG tracing during OW rowing would provide an opportunity to evaluate and monitor HR and rhythm of rowers in contexts consistent with naturally occurring demands. Despite the potential of wearable technology to provide real-time ECG data in athletic populations, diagnostic accuracy, reproducibility, and confidence in interpretation remain areas of uncertainty.

The Plux ECG Sensor [Plux Wireless Biosignals. Available online https://www.pluxbiosignals.com/ (accessed on 1 May 2022)] is a single-channel, low-powered, miniature, low-cost physiological sensor, which when coupled to a BIOSIGNALSPLUX unit for data acquisition (referred to here as the PluxECG) can be used to monitor the cardiac electrophysiology of people during intense exercise outside a laboratory setting [9]. However, this device has not been previously assessed for monitoring athletes in ‘the field’ or tested for reliability in the sport of rowing. Therefore, this study was designed in two phases. The first phase aimed to investigate the accuracy and precision of a single channel ECG system, (PluxECG) compared to a gold-standard 12-lead ECG assessment with high fidelity in rowers in a controlled laboratory setting. The second phase aimed to evaluate the confidence and reliability in the interpretation of ECGs recorded using the PluxECG device during remote rowing.

## 2. Materials and Methods

### 2.1. Study Design and Participants

The two phases of this study were: Phase I: assessment of the accuracy and precision of a single channel ECG system (the PluxECG);Phase II: evaluation of the confidence and reliability of remote ECG interpretation.

The protocol was approved by the Faculty of Health Sciences Research Ethics Committee, Trinity College Dublin.

Five healthy amateur female rowers (Phase I) were studied in Massachusetts General Hospital, Boston, Massachusetts, and twenty competitive male and female rowers (Phase II) were recruited from Rowing Clubs in Ireland. Participant inclusion criteria were age ≥ 18 years, rowing for a minimum of one year and currently training regularly (≥5 days/week), classified as a ‘senior club’ standard rower, free from current musculoskeletal disorders and no prior diagnosis of cardiac disease or rhythm disorder. Participants provided written informed consent and gave permission for their data to be published in accordance with the Declaration of Helsinki. 

### 2.2. Study Protocols and Data Collection

For both phases, participants were asked to refrain from ingesting caffeine and alcohol, and to avoid exercise or strenuous physical activity for 24 h. During each visit, a laboratory technician trained in ECG testing fitted participants with the PluxECG device [https://staging.biosignalsplux.com/downloads/docs/manuals/Electrocardiography_(ECG)_User_Manual.pdf (accessed on 10 January 2021)]. Prior to attachment of devices, the chest region of participants was shaved, cleaned, abrased and dried. The PluxECG device electrodes were positioned on the right sternum, left sternum, and mid-axillary line. Athletes were fitted with a polar heart rate strap and watch for their own use during testing.

#### 2.2.1. Phase I

Participants (*n* = 5) attended the Cardiac Performance Program laboratory of the hospital for exercise testing on two occasions, separated by at least 48 h. Participants were fitted with the PluxECG device and a reference 12-lead ECG system; MGC Ultima CPX™ metabolic stress testing system with Mortara 12-lead ECG ad on, the gold standard for exercise ECG testing [https://mgcdiagnostics.com/products/ultima-cpx-metabolic-stress-testing-system (accessed on 15 May 2021)]. Participants completed the same exercise testing protocol on both visits. Full details are outlined in the Appendix A. Briefly, each exercise test was divided into two different phases: Step Protocol and Ramp Protocol, which included rest and exercise at different levels of rowing intensity to maximise HR. HR data from the PluxECG and lab-based 12-lead ECG assessment was simultaneously recorded. Data processing and synchronisation of data from the PluxECG device and lab-based 12-lead ECG assessment is outlined in the Appendix A.

#### 2.2.2. Phase II

Demographic and anthropometric data, including age, sex, weight, and height were collected. Participants (*n* = 20) completed an incremental 2000 m (m) race simulation test under two conditions, a laboratory environment on the rowing ERG (Concept2 Model D, Morrisville, VT, USA) and an OW row with a single scull boat to determine the accuracy of the PluxECG device during high and moderate intensity exercise. For each participant the ERG test preceded the OW test and was completed on different days separated by at least 48 h.

The testing protocol was the ‘7 × 4 min (min) Step Test’ adopted by Rowing Australia [10] and remained the same for both ERG and OW testing conditions. For the ERG test, participants starting workload and incremental load were determined during their pre-assessment using their previous best 2000 m row time. Subsequently, the ERG drag was adjusted to the appropriate competition category for each participant, ranging between 140 and 200 watts. Following a 10 min warm-up on the participant’s preferred pace, 7 × 4 min increments were rowed. Each of the seven stages was separated by a one-minute recovery period. During the OW test participants were encouraged to use the Polar HR monitor as a surrogate reference to achieve similar effort as to their ERG test and were provided with target zones for each step. Step 2 and 7 from the ‘7 × 4 min Step Test’ were used as the reference points for moderate- and high-intensity exercise. The OW tests were conducted under fair weather conditions during spring and summer. Wind speed as measured by a Kestrel 1000 pocket wind meter (Nielsen-Kellerman, Boothwyn, PA, USA), was determined to be below 10 kilometres per hour. 

To determine agreement, 40 ECGs (20 ERG and 20 OW) from two rowing intensity levels (moderate and high; middle of the testing and at the end stage) were independently interpreted by two expert readers (cardiologists trained in athlete-specific ECG interpretation). Both readers were blinded to ECG testing conditions. HR were determined using R-R intervals, which were subsequently converted into instantaneous HR. ECGs were evaluated based on rhythm (*sinus*, a regular rhythm at any rate; *sinus with intermittent ectopic beats*, regular rhythm with intermittent atrial or ventricular ectopic beats; or *uninterpretable*, identification of rhythm was not possible), quality of tracing (movement artifact) and level of confidence in interpretation. For the quality of tracing and confidence in interpretation, readers were asked to assign a value using a five-point Likert scale. For quality of tracing: 1, excellent, no artifact; 2, minimal artifact; 3, an artifact with some interpretable complexes; 4, most of strip is an artifact; and 5, completely uninterpretable. For level of confidence in interpretation: 1, very confident; 2, fairly confident; 3, somewhat confident; 4, slightly confident; and 5, not confident. To determine reliability in ECGs between testing conditions (ERG and OW) and rowing intensity (moderate and high) expert readers consulted to establish consensus for an agreed final interpretation for each ECG.

### 2.3. Statistical Analysis

Analyses for Phase I assessed the agreement and correlation between the PluxECG device and the lab-based 12-lead ECG assessment. The agreement was estimated using the Bland–Altman method, adapted to consider repeated measures from the same participant when the true value varies over time [10,11]. The relationship between the paired differences and their averages were modelled to assess the extent to which the agreement varied for HR and estimated the intraclass correlation coefficient as an alternative measure of agreement. Correlation analysis was employed to quantify the linear relationship between the two devices, with the correlation coefficient providing a measure of the strength and direction of that relationship. Pearson correlation was used to compare HR measurements between the two devices for the Ramp Protocol. Correlation coefficient was determined using native MATLAB functions cov() and corrcoef(). A custom MATLAB script was developed to down-sample the PluxECG data due to being sampled at a higher rate than the lab-based 12-lead ECG, and thus aid comparisons of waveforms from both systems (Appendix A). 

Analyses for Phase II were performed using SPSS 22.0 for Windows (Chicago, IL, USA). Quantitative data is presented as mean ± standard deviation (SD) or median (interquartile range [IQR]). ECGs were classified according to rhythm interpretation (sinus, sinus with intermittent ectopic beats or uninterpretable) and are expressed as a percentage. Qualitative data, including quality of tracing “artifact level” and level of confidence in interpretation was evaluated using a five-point Likert scale. Differences between readers’ interpretation of ECGs according to condition (ERG vs. OW) and rowing intensity (moderate vs. high) were compared using an independent samples *t*-test for continuous data and Fisher’s exact test for ordinal data. The overall agreeability in ECG interpretation between readers for each condition was determined using the κ statistic; intraclass correlation for continuous data and Cohen’s kappa for ordinal data. Using final interpretation of ECGs, following the reaching of a consensus between the two expert readers, ECGs were rated to determine quality and reliability in interpretation according to condition and intensity. Quality of ECG ratings were determined using consensus findings between readers on the presence of artifacts and confidence in interpretation. Rating “++” indicates a highly positive rating (excellent, no artifact and very confident); “+” indicates positive reading (minimal artifact and fairly confident); “+/−”; indicates a neutral rating (artifact with some interpretable complexes and somewhat confident); “−” indicates poor rating (most of strip is an artifact and slightly confident); “−−” indicates very poor rating * completely uninterpretable and not confident). Differences in reliability between conditions for ECG interpretation was determined using the κ statistic. A *p*-value less than 0.05 was considered significant.

## 3. Results

### 3.1. Phase I

#### Agreement and Correlation between PluxECG Device and Lab-Based 12-Lead ECG

Bland–Altman analysis demonstrated that for instantaneous value of changing HR, 91.9% of the samples were within the 95% confidence interval, establishing the estimate of agreement between the PluxECG device and the lab-based 12-lead ECG system (Figure 1). 

The time difference between each PluxECG sample and the closest lab-based 12-lead ECG system sample in time was considered error with a mean of 0.1182 s (σ = 0.0066 s, SD = 0.0814 s). Visual inspection and manual alignment identified a clear trend between the data supplied by both devices (Figure 2).

For each participant and exercise test, the covariance and correlation between the PluxECG and the lab-based 12-lead ECG assessment data was calculated with a threshold HR set at 120 bpm (Appendix A). The mean correlation coefficient across participants and tests was found to be 0.9886 (σ = 0.0002 SD = 0.017). The mean correlation coefficient between the two devices for all participants at elevated HR was 0.9676 (σ = 0.002 SD = 0.05).

### 3.2. Phase II

The mean age of participants in Phase II was 32.4 ± 14.3 years and 80% (n = 16) were male. Mean height and weight were 182 ± 8.7 cm and 79.7 ± 12 kg, respectively.

#### Inter-Reader Reliability of PluxECG Device

Readers’ interpretations of ECGs at moderate- and high-intensity rowing under both ERG and OW conditions were significantly different (*p <* 0.05) for the presence of artifact and confidence in interpretation (Table 1). The level of agreement for ERG ECG interpretation was strong for HR (moderate-intensity: k = 0.93, *p* = 0.00; high-intensity: k = 0.95, *p* = 0.00) and weak for rhythm (moderate-intensity: k = 0.31, *p* = 0.00; high-intensity: k = 0.22, *p* = 0.02) and confidence in interpretation (moderate-intensity: k = 0.37, *p* = 0.00) (Table 2). For OW ECGs interpretation the level of agreement was moderate for HR (high-intensity: k = 0.70, *p* = 0.00) and weak for confidence in interpretation (high-intensity: k = 0.36, *p* = 0.00) (Table 2).

### 3.3. Differences between ECGs According to Testing Condition and Intensity

Using consensus data between readers for all participants ECGs, differences between HR, rhythm, artifact, and confidence in interpretation were explored for the moderate- and high-intensity ERG and OW rows. At moderate-intensity rowing, no significant differences were identified for all measures between ERG and OW testing conditions (Table 3). At high-intensity rowing, HR was lower for the OW row compared to ERG (*p* = 0.05). The presence of artifact was significantly different between ERG and OW at high-intensity rowing with ERG ECGs having a higher number of ECGs with greater levels of artifact than OW (*p* = 0.01).

### 3.4. Quality Ratings of ECGs 

For moderate-intensity rowing, ERG ECGs were more positively rated due to ECGs having less artifact and greater levels of confidence in interpretation than OW ECGs. At high-intensity rowing, ERG ECGs were less positively rated than OW ECGs due greater presence of artifact and lower levels of confidence in interpretation (Table 4).

## 4. Discussion

This is the first study to investigate the validity and reliability of the PluxECG device in an OW rowing environment (Phase I) and to evaluate the confidence and reliability in interpretation of ECGs recorded during laboratory and remote rowing (Phase II). The main findings in this study are that (1) a single channel ambulatory ECG system (PluxECG) records accurate and reliable HR data even at high-intensity exercise in rowers compared to gold standard methods of assessment; (2) the confidence in ECG interpretation using the PluxECG device in rowing athletes is inadequate at this time; (3) interpretation of ECGs for OW rowing have a lower level of reader agreement than ERG rowing; and (4) at high-intensity rowing, significant differences were apparent between OW and ERG rowing for HR and artifact.

Findings from Phase I indicate the PluxECG records accurate and reliable HR data at varying levels of exercise intensity compared to the gold standard method of assessment. The 12-lead ECG registers each electrical impulse of the heart for the quantification of R-R intervals and is the current gold standard method in ambulatory settings [12,13,14]. Pearson correlations and Bland–Altman analyses reflected a very high agreement between the PluxECG device and the lab-based 12-lead ECG system. Regarding the Pearson correlations, strong positive relationships were identified between the participants tests and between the two systems at elevated HR (Appendix A). The Bland–Altman analysis plots revealed that 92% of the samples fell within the 95% confidence interval for instantaneous value of the changing HR (Figure 1). The agreement between devices is further supported through the small median differences identified between the two systems’ intervals and rate of change of HR; this illustrates that the PluxECG device provides accurate and precise HR data during high-intensity exercise, such as rowing. 

Findings from Phase II of this study indicate that using the PluxECG device as an instrument for ECG interpretation is challenging. Poor levels of agreement between readers for ECG tracing, both of whom are very experienced in ECG interpretation, suggest that the PluxECG device is not optimised for electrocardiographic interpretation for diagnostic accuracy. Clinicians’ confidence in ECG interpretation using PluxECG device was low, primarily attributed to artifact. Unlike measuring R-R intervals, used for assessing HR, reliable ECG interpretation requires the accurate transmission of high-quality ECGs without loss of detail, including, *p* waves, QRS axis, and duration of the PR and QRS segments and intervals. Commonly, ECG devices used during exercise are contaminated by several artifact types, including respiration, body movement, and muscle contractions [15]. Portable ECG monitors typically utilise lower cost sensors and reduced number of leads (1- and 3-lead), resulting in increased sensitivity to such artifacts, particularly due to movement, thus hampering signal quality. Additional analysis from Phase I indicated that the PluxECG device generated considerable movement artifact (Appendix A). Artifacts related to portable ECG devices were originally classified as pseudo-arrhythmias and non-arrhythmia artifacts [16] and are most probably related to body movement, temporary impairment of skin-electrode contact, loose electrode connections, broken leads, skeletal myopotentials, and ambient noise [17]. Noise and interferences have the potential to hide essential parameters that are necessary for reliable visual interpretation. High levels of artifact reduce a clinician’s ability to interpret the presence of arrhythmias or complex rhythms, increasing the potential for erroneous interpretation. 

Athletes sometimes experience transient arrhythmias during periods of intense sport-specific exercise [4,18], which can be difficult to emulate in laboratory settings using traditional Holter monitors. Distinct differences in movement dynamics and cardio-respiratory demands between the ERG and OW rowing have been documented [19]. Energy system contribution comparisons indicate a higher aerobic demand for on-water rowing [20], which raises concerns on the reproducibility of cardiac characteristics of rowing performance between the two conditions. Therefore, the ideal use of the small wearable devices (such as the PluxECG) would be to obtain ECG tracing during OW rowing. Our findings show clear inconsistencies in the quality of data from the PluxECG device regardless of testing environment (ERG or OW rowing) and intensity (Table 4). Research on validation of ECG devices to evaluate exercise-related arrythmias in athletes is sparse despite their widespread and growing use. Recent systematic reviews assessed wearable devices to monitor cardiac health and ECG traces in the general population [21,22]. However, these findings cannot be directly applied to athletic populations as ECGs are assessed at baseline and not during high-intensity exercise where movement artifact and abnormal waveforms occur. Although many devices are available, only three have received medical device regulatory approval to discriminate between sinus rhythm and AF; these include the Apple Watch, Fitbit and AliveCor Kardia system. The Kardia device has been found to accurately detect arrythmias in athletes in a case series [23] and deemed to have good reliability compared to a 12-lead ECG device [24]. The Apple Watch (series 4 and newer) has been shown to have agreement with a standard 12-lead ECG device for the classification of the rhythm as sinus or AF. However, its use is in athletes is limited as activity must be paused for the recording, contradicting the idea of measurement during training [25]. 

There are some limitations in this study. The PluxECG and 12-lead ECGs were obtained contemporaneously but temporal synchronisation was not possible. However, this does not impact the analysis of variability in HR or rate of change of HR measurements. The inability to extract raw ECG data beyond HR from the lab-based 12-lead ECG assessment limited the ability to assess the reliability of the PluxECCG device beyond HR during Phase I. Due to the recognised difficulties of obtaining a lab-based 12-lead ECG assessment, ‘gold standard’ during field testing, such as OW rowing we were unable to compare the accuracy of the PluxECG device. Further investigation on the PluxECG device is warrant due to the modest sample size included, which limits our ability to draw definite conclusions. 

## 5. Conclusions

A single channel ambulatory ECG system (PluxECG) records accurate and reliable HR data even at high-intensity exercise in rowers; this indicates that this system can be used to provide a reliable measure of HR during training. However, using the PluxECG device as an instrument for ECG interpretation during sport-specific exercise in athletes is challenging. Inconsistencies in clinicians’ interpretation and quality of ECG tracing were apparent, irrespective of exercise conditions and intensity. Portable devices used in athletic populations need to produce reliable and accurate ECGs during sport-specific training. There remain several barriers to overcome if such technology is to become the contemporary standard in arrhythmia detection in athletes. Current iterations of single lead ECG devices are reliant on HR monitoring, and R-R intervals for rhythm determination but are sensitive to interference, such as movement artifact, hampering the reliability in interpretation. There is limited research assessing the validity of such devices in athletes during high-intensity exercise and sport-specific training even though they are marketed for use in such settings. Further evaluation of the wearable ECG devices should include assessment of interval measurements beyond R-R intervals to determine the accuracy of ECG tracing compared to gold standard methods of assessment.

## Figures and Tables

**Figure 1 diagnostics-12-02271-f001:**
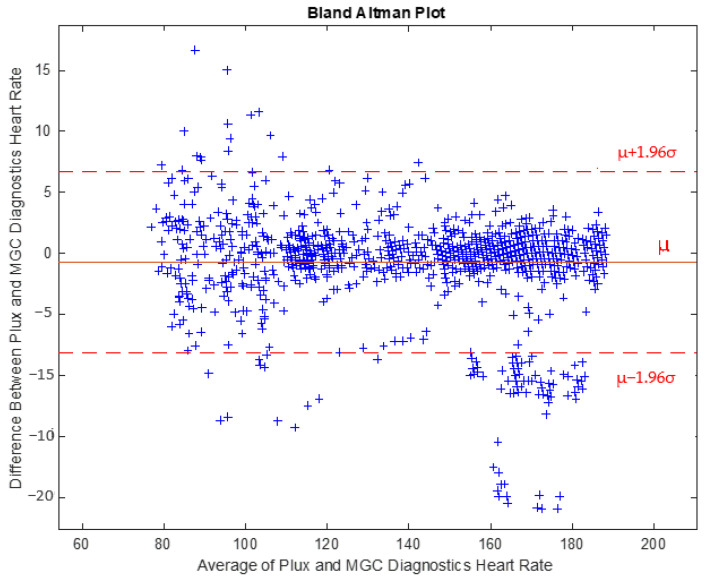
Bland Altman Bland–Altman Plot indicating mean difference in HR detection between the PluxECG and the lab-based 12-lead ECG system measurement. Mean bias and Limits of Agreement (LOA) (95% LoA) are shown.

**Figure 2 diagnostics-12-02271-f002:**
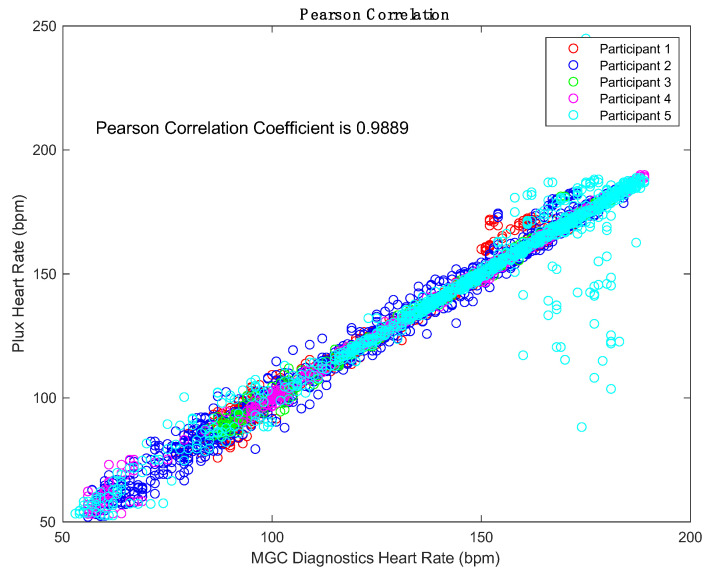
Pearson Correlation comparing PluxECG and the lab-based 12-lead ECG system heart rate measurement for the Ramp Protocol.

**Table 1 diagnostics-12-02271-t001:** Comparisons between readers on ECG interpretations by rowing intensity and condition.

	Ergometer Row			On-Water Row		
	Reader 1	Reader 2	*p*-Value	Reader 1	Reader 2	*p*-Value
Moderate-intensity row						
Heart rate, mean ± SD	147.6 ± 37.2	147.4 ± 32.5	0.99	148.5 ± 41.7	153.3 ± 42.2	0.72
Sinus rhythm, n (%)	18 (90%)	19 (95%)	0.10	19 (95%)	18 (90%)	1.00
Artifact, median (IQR)	2 (1)	2 (1)	0.01	2 (1)	3 (1)	0.31
Confidence in interpretation, median (IQR)	2 (1)	2 (2)	0.02	2 (0)	3 (2)	0.05
High-intensity row						
Heart rate, mean ± SD	187.2 ± 45.8	183.5 ± 39	0.79	148.5 ± 46.7	159.7 ± 28.7	0.37
Sinus rhythm, n (%)	17 (85%)	19 (95%)	0.06	19 (95%)	19 (95%)	1.00
Artifact, median (IQR)	3 (2)	4 (1)	0.03	2 (2)	3 (2)	0.03
Confidence in interpretation, median (IQR)	2 (1)	4 (2)	0.01	2 (1)	2.5 (3)	0.00

Continuous data is presented as mean ± SD. Nominal data is presented as n (%). Ordinal data is presented as median (IQR). Differences between readers were established using an independent samples *t*-test for continuous data and Fisher’s exact test for ordinal data. *p <* 0.05 is significant.

**Table 2 diagnostics-12-02271-t002:** Inter-rater reliability between readers.

	Ergometer Row				On-Water Row			
	Moderate-Intensity Row		High-Intensity Row		Moderate-Intensity Row		High-Intensity Row	
	k	*p*-Value	k	*p*-Value	k	*p*-Value	k	*p*-Value
Heart rate	0.93	0.00	0.95	0.00	0.37	0.16	0.70	0.00
Rhythm	0.31	0.00	0.22	0.02	0.03	0.73	−0.03	0.81
Artifact	0.16	0.21	−0.05	0.63	0.07	0.41	0.07	0.51
Confidence in interpretation	0.37	0.00	0.03	0.74	0.07	0.40	0.36	0.00

All values represent Intraclass Correlation—Cronbach’s Alpha for scale data and Cohen’s kappa value of ordinal data.

**Table 3 diagnostics-12-02271-t003:** Differences between Erg and OW ECGs for moderate and high-intensity exercise using consensus data from ECG interpretations.

	Moderate-Intensity Row			High-Intensity Row		
	ERG ECG (n = 20)	OW ECG (n = 20)	*p*-Value	ERG ECG (n = 20)	OW ECG (n = 20)	*p*-Value
Heart rate	145.6 ± 39.3	150.6 ± 42.1	0.70	181.7 ± 46.3	142.5 ± 45.8	0.05
Rhythm			1.00			1.00
1	19 (95%)	19 (95%)		18 (90%)	18 (90%)	
2	-	-		1 (5%)-	1 (5%)	
3	1 (5%)	1 (5%)		1 (5%)	1 (5%)	
Artifact			0.51			0.01
1	-	-		-	1 (5%)	
2	12 (60%)	9 (45%)		3 (15%)	12 (60%)	
3	6 (30%)	9 (45%)		9 (45%)	3 (15%)	
4	1 (5%)	-		6 (30%)	2 (10%)	
5	1 (5%)	2 (10%)		2 (10%)	2 (10%)	
Confidence in interpretation			0.40			0.07
1	5 (25%)	1 (5%)		1 (5%)	7 (35%)	
2	8 (40%)	10 (50%)		5 (25%)	7 (35%)	
3	6 (30%)	6 (30%)		9 (45%)	3 (15%)	
4	-	2 (10%)		4 (20%)	2 (10%)	
5	1 (5%)	1 (5%)		1 (5%)	1 (5%)	

Continuous data is presented as mean ± SD. Nominal data is presented as n (%). Ordinal data is presented as median (IQR). *p <* 0.05 is significant. Rhythm: 1, sinus; 2, sinus with intermittent ectopic beats; and 3, uninterpretable. Artifact level: 1, excellent, no artifact; 2, minimal artifact; 3, artifact with some interpretable complexes; 4, most of strip is artifact; and 5, completely uninterpretable. Confidence in interpretation: 1, very confident; 2, fairly confident; 3, somewhat confident; 4, slightly confident; and 5, not confident.

**Table 4 diagnostics-12-02271-t004:** Quality rating of ECGs.

Moderate-Intensity Row				
Rating Scale	Ergometer ECG (n = 20)		On water ECG (n = 20)	
	Artifact	Confidence of ECG Interpretation	Artifact	Confidence of ECG Interpretation
++	0	5	0	1
+	12	8	9	10
+/−	6	6	9	6
−	1	0	0	2
−−	1	1	2	1
**High-intensity row**				
**Rating Scale**	**Ergometer ECG** **(n = 20)**		**On water ECG** **(n = 20)**	
	**Artifact**	**Confidence of ECG interpretation**	**Artifact**	**Confidence of ECG interpretation**
++	0	1	1	7
+	3	5	12	7
+/−	9	9	3	3
−	6	4	2	2
−−	2	1	2	1

Quality of ECG ratings were determined using consensus findings on the presence of artifact and confidence in interpretation. Rating “++” indicates a highly positive rating (excellent, no artifact and very confident); “+” indicates positive reading (minimal artifact and fairly confident); “+/−”; indicates neutral rating (artifact with some interpretable complexes and somewhat confident); “−” indicates poor rating (most of strip is artifact and slightly confident); “−−” indicates very poor rating (completely uninterpretable and not confident).

## Data Availability

Not applicable.

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
