# Peer review of "Diagnostic Accuracy of a Portable ECG Device in Rowing Athletes"

_diagnostics, 2022, doi:10.3390/diagnostics12102271_

Round 1

Reviewer 1 Report

The article Diagnostic Accuracy of a Portable ECG Device in Rowing Athletes presents a good method to evaluate portable devices used during sports training activity, with the idea that if this technology is to become a contemporary standard must produce reliable and accurate ECGs. Here are some suggestions to improve the article.

-I couldn't find a manufacturer for PluxECG, or a device maned PluxECG. Can you give us a direct link to data specs or some marketing data? So the reader can learn the technical specifications.

-Can you expand in more detail on what defines an expert reader? Also, can you give specs and precision for what an artifact would be?

-Reference (9) does not show anything about the PluxECG as the last paragraph of the Introduction feels

-You declare that the portable devices are “sensitive to interference” but you don’t describe the specific interference noticed. Give more details on the interference type.

-The lack of phase linearity in a bandpass filter can distort the signal. Can you express the phase quality of you filters?

-Figures 5 to 8 lack a narrative in the text that calls them

-Figure 8 lacks the subject number, or is this typical for all subjects?

-Can you suggest what a wearable device should produce to declare improved confidence in interpretation?

Reviewer 2 Report

In their study, Wilson and colleagues investigate the reliability of a portable single channel ECG device (PluxECG) and evaluate the confidence and reliability in interpretation of ECGs recorded using the same device during remote rowing. 

The methodological design is appropriate for assessing the diagnostic accuracy or confidence and reliability in interpretation of ECGs recorded of PluxECG in rowers. Noteworthy is the evaluation of the confidence and reliability of the device's interpretation during remote rowing, beyond conducting tests in a controlled laboratory environment.

One important limitation is, in my opinion, is the small sample size, especially in “phase I” which included only 5 female participants. 

Some minor comments to the manuscript:

The PluxECG (SENSPRO-ECG1-30-30-30) is unobtrusive single-lead ECG data acquisition sensor (BIOSIGNALSPLUX), but for data acquisition did the authors use biosignalsplux acquisition system or did they develop their own acquisition system? Please add to the manuscript.

Time stamped HR data from the MGC Diagnostics system was stored in a unique logfile (.txt) for each test and each participant. However, there is no mention of how the ECG signal data was sent or stored using the Plux-ECG. Please add to the manuscript.

Reviewer 3 Report

 The heart rate and ECG from a portable PluxECG were evaluated on rowing athletes. The idea of testing portable monitoring device on specific type of athletes is interesting and worth investigation. The methods of experiments are reasonable and sound. However, the outcome of the study is not very encouraging, and measurement of ECG was not reliable with PluxECG. The reviewer recommends major revision to improve the quality of the manuscript. These comments must be addressed before publication.

1-      The results show that PluxECG is effective for measuring heart rate. Can heart rate be measured with some other wearable devices such as PPG sensors? The authors should comment why PluxECG can be useful for heart rate.

2-      The Introduction of the manuscript needs more explanation about other methods for measuring ECG (e.g., Forth Frontier X2) especially for athletes. Some parts of discussion about Apple Watch etc.) can be included in the introduction.

3-      The authors must comment about the status of electrodes during high intensity rowing. Were they secure?

4-      The manuscript mentioned that “custom MATLAB script was developed to down-sample the PluxECG data…”. This down sampling may affect the motion artifacts on the PluxECG. The comparison with 12-lead ECG may not be accurate.

5-      The results show that PluxECG is not effective for measuring ECG on rowing athletes. The authors may consider using other types of wearable ECG devices for this sport and compare the results.

Round 2

Reviewer 3 Report

The responses to comments are reasonable.

The responses could be implemented in the manuscript.